# Plants Utilize Suberin Biopolymers as a Vector for Transmitting Visible Light through Their Roots

**DOI:** 10.3390/polym14245387

**Published:** 2022-12-09

**Authors:** Spenser Waller, Stacy L. Wilder, Michael J. Schueller, Richard A. Ferrieri

**Affiliations:** 1Missouri Research Reactor Center, University of Missouri, Columbia, MO 65211, USA; 2School of Natural Resources, University of Missouri, Columbia, MO 65211, USA; 3Chemistry Department, University of Missouri, Columbia, MO 65211, USA; 4Interdisciplinary Plant Group, University of Missouri, Columbia, MO 65211, USA

**Keywords:** plant light conduction, root endodermis, suberin biopolymers

## Abstract

Plants conduct light from their aboveground tissues belowground to their root system. This phenomenon may influence root growth and perhaps serve to stimulate natural biological functions of the microorganisms associating with them. Here we show that light transmission in maize roots largely occurs within the endodermis, a region rich in suberin polyester biopolymers. Using cork as a natural resource rich in suberin polymers, we extracted, depolymerized, and examined light transmission in the visible and infrared regions. Suberin co-monomers dissolved in toluene showed no evidence of enhanced light transmission over that of the pure solvent in the visible light region and reduced light transmission in the infrared region. However, when these co-monomers were catalytically repolymerized using Bi(OTf)_3_, light transmission through suspended polymers significantly increased 1.3-fold in the visible light region over that in pure toluene, but was reduced in the infrared region.

## 1. Introduction

In nature, the root system of higher plants is mostly embedded in soil and thus assumed to be blocked from light exposure. On the other hand, the leaves and shoots growing aboveground are exposed to sunlight. These tissues are known to possess various light-sensing molecules which mediate photosynthesis and photomorphogenesis. However, recent accumulating evidence strongly support that the roots are also capable of sensing and responding to light which can trigger root morphogenic dynamics and developmental changes, which mediate primary root growth, lateral root formation, nutrient uptake, and greening process [1,2,3]. In fact, genes encoding photoreceptors have been shown to be expressed in root cells [4,5] and can be activated by direct light stimulation [6,7,8,9]. Normally, sunlight striking the surface of the soil will penetrate only a few millimeters in depth [10]. Hence, conduction of light to the roots by this means will be highly variable and dependent on both soil composition and layering [10,11]. Alternatively, light could be conducted through plant tissues directly to the roots providing greater penetrability [12,13].

Aside from the growth promoting benefits of light conduction belowground as perceived by changes in root architecture, there may be additional benefits regarding how light conduction might stimulate the microbiome. Many biological activities and physiological processes that occur in nature with a periodicity of around 24 h have been described in many organisms [14]. Circadian cycles, synchronized and entrained by light and dark periodicity can often determine behavioral patterns, particularly in mammals such as cycles of sleep and consciousness. In plants, the production of hormones, the onset of certain developmental processes, or the release of seed and root exudates can be subject to the same periodicity [15]. Over the years there has been a mind-set in the scientific community that such periodicity in biological functions is absent for most microorganisms in the soil microbiome, except in the case of cyanobacteria, in which photosynthesis and nitrogen fixation are usually temporally separated and controlled. Even so, there is mounting evidence from molecular sequencing that shows genes encoding phytochrome-like proteins are present in the genomes of many heterotrophic bacteria that can sense light and synchronize periodic behavior to light availability [16,17,18,19,20].

Within the betaproteobacteria, the genus *Herbaspirillum* is heterotrophic and comprises several, mostly diazotrophic plant growth promoting species [21], some of which exhibit the potential of endophytic and systemic colonization across the plant kingdom [22,23]. In the past, root-colonization by *Herbaspirillum seropedicae* has been detected both on root surfaces and as endophytes within the intercellular spaces, as well as within intact root cells [24,25,26,27]. 

Because endophytic microorganisms depend on their host for nutrients to support their energy demands and ATP synthesis [27,28], microbial metabolic functions should be affected by variations in the physiological properties of their hosts, such as leaf photosynthesis whose rate influences the flux of photosynthates belowground. This is especially true of diazotrophs whose ability to fix N_2_ is an energy taxing process requiring 16 ATP molecules for every N_2_ reduced by the organism. However, plant photosynthesis during the light cycle might increase O_2_ concentrations around the endophyte thereby repressing expression of its nif structural genes that are responsible for regulating nitrogenase and N_2_-fixation. Even so, past research has shown that levels of transcription factors associated with nif gene expression were significantly elevated during the host’s light cycle [29]. This raises the important question whether light conduction within the plant roots could influence the biological functions of the microorganisms colonizing these tissues.

In the present study, we sought to examine light transmission in maize roots and the suberin biopolymers therein, to gain insight into the physicochemical basis for why this phenomenon occurs. What led us to examine the suberin polyester specifically, and its potential light transmission properties was because our earlier published work examining the influences of certain maize root-associating bacteria on host physiological and metabolic responses [30] led to the observation that such associations can cause a thickening of the Casparian strips within the root endodermis. This morphological change was due to increased suberization within the cell wall apoplastic space. This hydrophobic barrier acts as a regulation point controlling the movement of micronutrients into the vascular core of the plant. 

Suberin is a highly complex biopolymer made up of long chain fatty acids ranging in size from C_16_–C_24_ in length, glycerol molecules and other aromatic substituents [31,32]. Within the apoplastic space occupying plant cell interfaces, a series of complex enzymatic steps contribute to the fusing of these monomeric building blocks to form the biopolymeric matrix defined as suberin [33]. Implicit in the highly plastic nature of suberin formation, matrix composition can vary both in form and function across different tissue types within a single plant species, as well as across the plant kingdom [34].

Recent work examining suberin from a cell biology perspective has enriched our knowledge of the cellular trafficking of its monomeric components and their polymerization all contributing to our fundamental understanding of suberin biosynthesis, regulation, and biological function [34]. Even so, there is much to be learned about the physicochemical properties of this complex matrix and its role in sustaining life processes.

## 2. Materials and Methods

### 2.1. Visualizing Light Transmission in Maize Roots

A digital camera was mounted inside a dark box (Figure 1) which allowed for the lower stem portion of a 3-week-old maize plant to protrude into the dark box. An isolated root was cut using a razor blade and positioned in the focal plane of the macro lens using a micro translation plate. The upper stem of the plant was cut and fitted into one side of a light tight metal tube positioned into the wall of the dark box. The other side of the tube was fitted to a fiber optic halogen lamp (Amscope 150 W fiber optic microscope illuminator, Microscope Central, Inc., Feasterville, PA, USA) with light output from 400 nm to 1200 nm.

An image intensifier was used for these studies which was manufactured by Sofradir (EC9350-CCD-PRO) and marketed under the name Astroscope (B&H Photo, Inc., New York City, NY, USA). This tube was configured to operate when affixed to a DSLR camera (Canon EOS 5D Mark III) and was installed between the lens (Canon MP-E 65 mm f/2.8 1-5x macro lens) and the camera (B&H Photo, Inc., New York City, NY, USA). The photocathode was the GaAs (Gen III) type sensitive to light mapping from 350 nm to 900 nm. The intensifier turns light to electrons, multiplies them while maintaining 2D spatial position with the aid of a microchannel plate (Figure 2). These electrons strike a phosphor screen giving off an amplified signal as light that the camera sensor records.

### 2.2. Depolymerization of Suberin

Here we followed published procedures for performing alkaline hydrolysis of cork suberin yielding a source of suberin co-monomers [35,36]. All chemicals and reagents used in the procedure were purchased from Sigma-Aldrich, Inc. (St. Louis, MO, USA). The suberin depolymerization products were isolated from industrial cork powder samples, which was subjected to an alkaline hydrolysis depolymerization procedure. Briefly, alkaline hydrolysis was carried out with a 0.5 m NaOH solution in ethanol/water (2.8 g NaOH dissolved in 90 mL ethanol/water, 9:1 *v*/*v*) at 70 °C for 1.5 h. The ensuing mixture of hydrolyzed suberin co-monomers was cooled to room temperature, acidified using dilute HCl to pH 3–3.5 and extracted three times using dichloromethane. The dichloro methane solvent was evaporated in a rotary evaporator and the resultant film scraped from the flask, weighed, and resuspended in toluene for light transmission measurements. The aliphatic suberin extract, obtained by the hydrolysis of its crosslinked natural precursor, is mainly composed of C_16_–C_24_ ω-hydroxyalkanoic acids and α, ω-alkanedioic acids (Figure 3), or the corresponding methyl esters. However, we would expect little or no epoxide monomer owing to the alkaline hydrolysis cleaving the epoxide ring [36].

### 2.3. Catalytic Polymerization of Suberin Comonomers

For repolymerization of cork-derived suberin comonomers we followed published procedures [37] to yield the macromolecular polyester structures shown in Figure 4. Typically, reactions were carried out using 0.5 g of comonomers mixed in 100 μL glycerol and 0.0213 g of bismuth (III) trifluoromethanesulfonate (Bi(OTf)_3_). The contents were stirred into 2 mL of 1,4-dioxane, heated to 90 °C, and stirred for 1 h. Vacuum was gradually applied over 48 h to remove water and 1,4-dioxane. The remaining contents were dissolved in 25 mL dichloromethane and the polymer was precipitated by pouring the contents into 1 L of cold methanol. This step removed the Bi(OTf)_3_ catalyst and other soluble oligomers. The precipitate was filtered and dried under vacuum at 40 °C for 24 h.

Validation that co-monomers underwent a chemical transition to a polymerized state was carried out by a simple melting point determination (Thermo Scientific Fisher-Johns melting point apparatus, Cole-Palmer, Inc., Vernon Hills, IL, USA). The co-monomer sample melted over a range of 25–38 °C while the polymerized sample melted over a higher temperature range from 42–53 °C. These melting ranges matched with prior published results for cork derived suberin [37].

### 2.4. Quantifying Light Transmission

Solutions of suberin co-monomers and polymer were prepared at 1 mM concentration in toluene solvent and dispensed into 300 mm × 12.7 mm o.d. Teflon tubes that were fitted with optical quartz windows. Each tube was wrapped in aluminum foil and placed in the dark box (Figure 1) configured for measuring light transmission through the tube using the halogen lamp (Figure 5).

Light registered by the camera sensor was quantified using ImageQuant™ TL 7.0 software (Cytiva Life Sciences, Inc., Marlborough, MA, USA) and recorded as the fractional light transmission with water having a value of 1.0.

### 2.5. Statistical Analysis

Data was analyzed using the student’s t-test for pair-wise comparisons made in light transmission measurements between pure toluene solvent and 0.5 mM solutions of suberin co-monomers and the co-polymer. Statistical significance was set at *p*-values less than 0.05.

## 3. Results

### 3.1. Visualizing Light Transmission in Maize Roots

Using the experimental setup depicted in Figure 2, we were able to observe light emitted from cut root ends of maize primarily within the endodermis (Figure 6). We note that this same region can be rich in endophytic bacteria like *H. serpedicae* [27]. 

### 3.2. Quantifying Light Transmission in Suberin

Using the experimental setup depicted in Figure 5, we quantified full light transmission through the suberin co-monomers (Figure 7A) and the repolymerized suberin (Figure 7B). Under the full light spectrum, the fractional light transmission through the suberin polymer solution was 1.3 ± 0.2, which was statistically higher than the 0.8 ± 0.1 transmission value found for the co-monomer. Light transmission through pure toluene in the full spectrum regime was identical to water. Contrary to this, infrared light transmission through both the co-monomer and the polymer were significantly less than that through pure toluene. 

## 4. Discussion

Higher plants have evolved the capacity to develop protective barriers as a first line of defense against environmental challenges from abiotic and biotic stressors. Such barriers can be achieved by the formation of hydrophobic biopolymers deposited at the periphery of cells. Suberin is one of such biopolymers comprised of polyester linkages that are deposited just beneath the plant’s primary cell wall. As a complex biopolymer found in both aboveground and belowground tissues, the plasticity of suberin’s formation and deposition is evident by how it is subject to environmental cues [34].

Within the plant’s root system, the endodermis has evolved with the development of two barriers: one comprised of the Casparian strips which establish an apoplastic barrier; the other the suberin lamellae that prevents diffusion through the plasma membrane [30]. Together these barriers work to regulate the movement of water, solutes, gases, and pathogens from the outer epidermal interface with the soil to the inner vascular core.

While the environmental framework defining suberin’s formation *in planta* is well characterized [34], we are still learning that this biopolymer can possess other unique properties, as evidenced by the results in the present work, that can benefit both plant and perhaps even the associating microorganisms growing within the roots. Here, we show conclusive evidence that light transmission from shoots-to-root in maize is conducted largely within the endodermal tissues occupied by suberin biopolymers. Admittedly, our extrapolation of observations on the light transmitting properties of cork derived suberin to the maize model might be limiting in some respects because of the varied nature in suberin chemical makeup across plant species [34]. Even so, light transmission in suberin was shown to have some unique properties. That is, this phenomenon occurs only in the visible light spectrum. That said, both the suberin co-monomers and polymers exhibited lower light transmission than water and toluene in the infrared region which may be attributed to increased absorption of infrared energy by the chemical bonds causing increased vibration and stretching of the bonds. We note that pure toluene solvent showed higher light transmission than water in the infrared which as an aromatic molecule offers very little opportunity for chemical bond excitation. At this point, we can say little about the mechanism for how light is transmitted through the polymeric material in the broad visible light spectrum of the halogen lamp. Perhaps the small amount of ultraviolet light that these lamps put out could electronically excite the polymeric material in some way. Perhaps, the polyaromatic domains incorporated into the suberin matrix contribute in some way to this mechanism of action. To further explore this phenomenon, we would need to have greater control over the spectral radiant energy input, as well as compare suberin derived from several different plant resources and not just cork, with a more detailed compliment of analytical workup characterizing the chemical properties of each. 

Finally, we suggest there could be unrealized benefits to light conduction in plant tissues resulting in stimulation of growth and performance of microorganisms such as plant growth promoting endophytes like *H. seropedicae*. Here too, we will seek to examine this possibility using fluorescent reporting strains of this bacteria and other root-associating bacteria to measure their growth performance *in planta* under different wavelengths of light. 

## Figures and Tables

**Figure 1 polymers-14-05387-f001:**
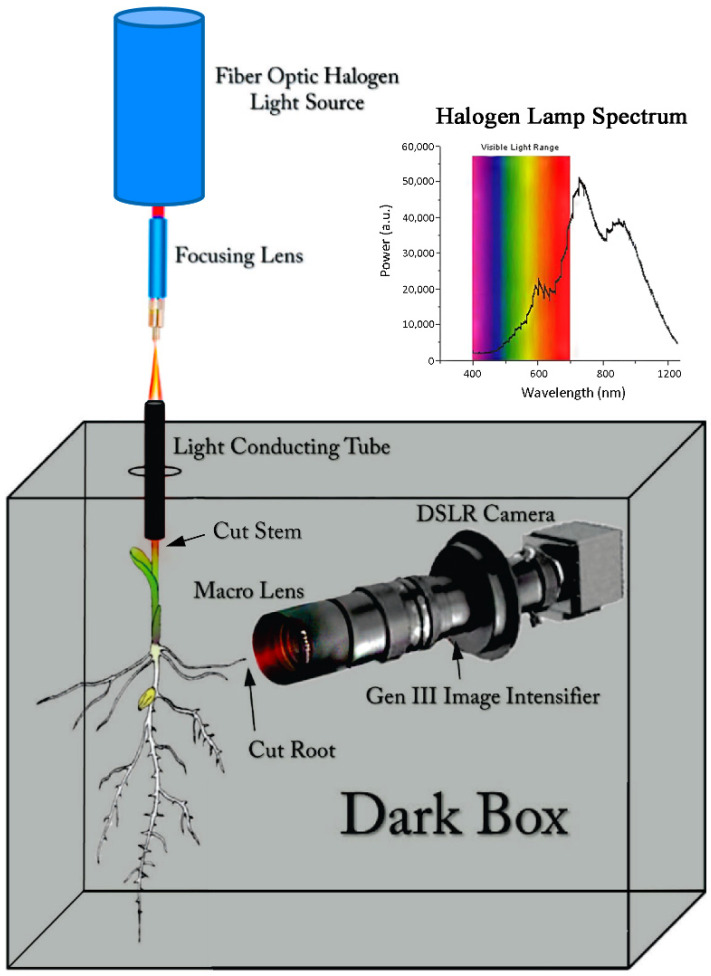
Experimental setup for visualizing light conductance through a maize plant’s stem to its root tip.

**Figure 2 polymers-14-05387-f002:**
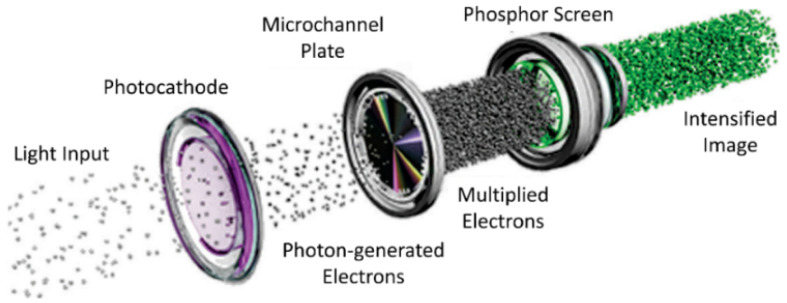
Schematic of the image intensifier tube.

**Figure 3 polymers-14-05387-f003:**
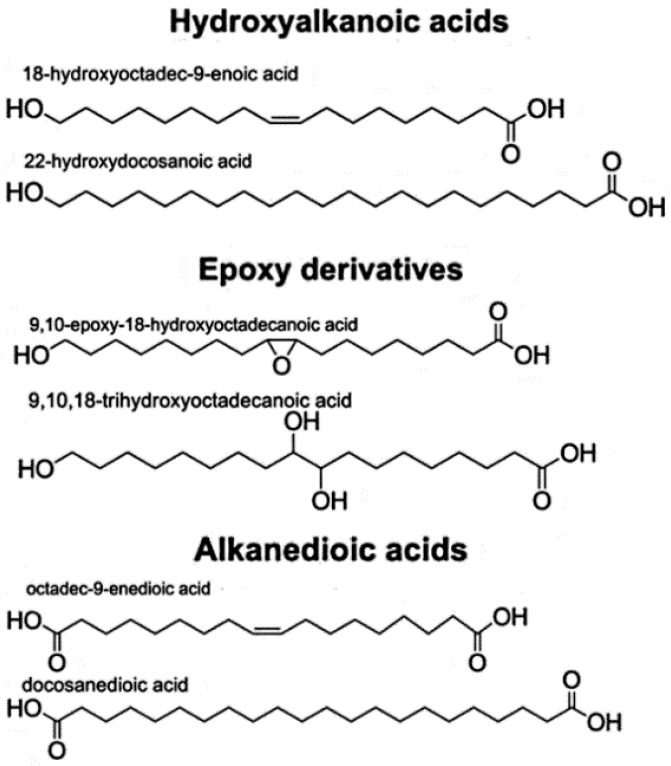
Structures of suberin co-monomers (Reprinted with permission from Ref. [36]. Copyright 2008, John Wiley & Sons Publishing, Wiley-VCH Verlag GmbH & Co. KGaA, Weinheim, Germany).

**Figure 4 polymers-14-05387-f004:**
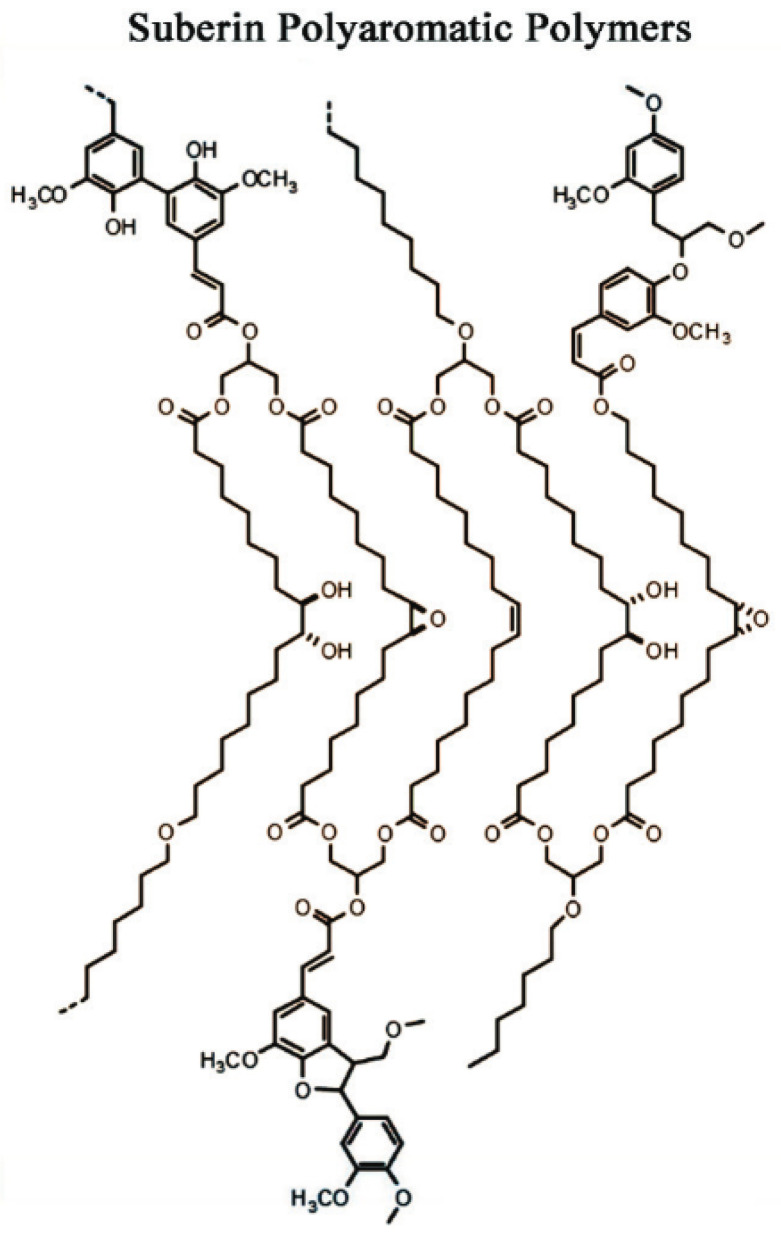
Macromolecular structure of suberin polyester.

**Figure 5 polymers-14-05387-f005:**
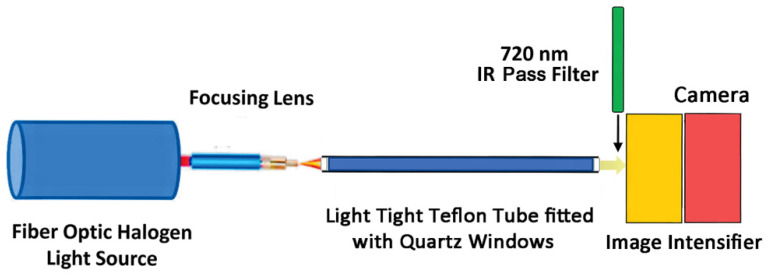
Experimental setup for measuring light transmission.

**Figure 6 polymers-14-05387-f006:**
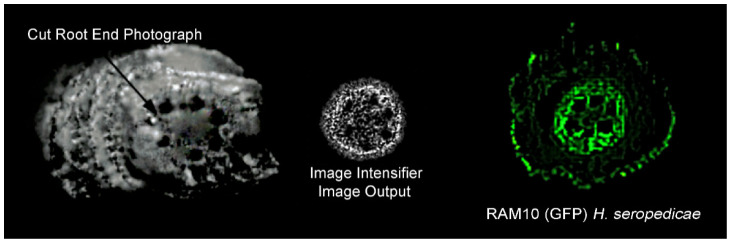
The macrophotograph of the cut root end shows xylem vasculature (darkened regions) in the center core. In the center is an image of the intensifier output of light showing higher light transmission in the endodermal ring surrounding the inner vascular core. The green fluorescence image on the right side was adapted from our prior studies using a GFP reporting strain of *H. seropedicae* [27] to identify regions within root transections that showed high levels of microbial colonization.

**Figure 7 polymers-14-05387-f007:**
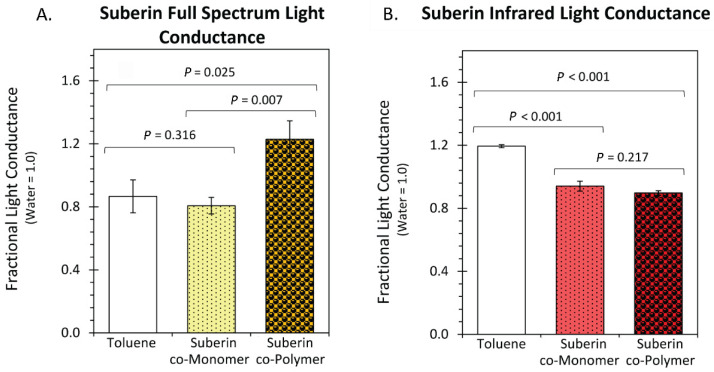
Light transmission graphs for full spectrum light Panel (**A**) and infrared light Panel (**B**) compare fractional transmission values for toluene solvent and 0.5 mM solutions of the suberin co-monomers and co-polymer. All data reflects average values ± SE for N = 6 replicates. *p*-values are shown in each panel showing levels of significance. *p* < 0.05 was considered statistically significant.

## Data Availability

Data will be provided on request to the corresponding author.

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
