# Peer review of "Plants Utilize Suberin Biopolymers as a Vector for Transmitting Visible Light through Their Roots"

_polymers, 2022, doi:10.3390/polym14245387_

Round 1

Reviewer 1 Report

I suggest more referencies within the text, and explanation of some biological terms in more details and how are they collerated with significance of understanding light transmission of plants. How this finding correlate with future development in biomaterials an generally undrstanding of relation between light transmittion and plant finctionallity.

Author Response

Responses to Reviewer 1 are listed below in italics.

I suggest more references within the text, and explanation of some biological terms in more details and how are they correlated with significance of understanding light transmission of plants. Please see Introduction (L79-92) where new text and new references (30-34) were added to give clarity to why we were looking at the light conduction properties of suberin biopolymer.  That text and included references also introduces the Casparian strip within the root endodermis.

How does this finding correlate with future development in biomaterials and generally an understanding of relation between light transmission and plant functionality. As a plant scientist any conclusions drawn regarding future developments of suberin-based biomaterials would be speculative at best. The conclusion does discuss light transmission in plants with possible benefits to the microorganisms within the microbiome.

L25     Please elaborate more about what are roots beyond the ground and what are on the ground in respect to shoots and other parts of plant root. Edits were made to lines 26-31 that should aid in clarifying the plant parts. 

L27      Please explain what is in this sense aerial light. I changed “aerial light” to sunlight.

L35     Add a reference here. Reference 10 which was cited on L38 was moved up and cited on L35.  That reference discusses sunlight penetration of soil.

L48     Do you refer to all microorganisms or to those assassinated with the plants? Please explain in the text. I added the line “microorganisms in the soil microbiome” which is inclusive of all microorganisms.

L82     I suggest that first goes Fig.2 which explains experimental set-up, and then Fig 1, which explains the image intensifier tube.  Figures 1& 2 were swapped in their order.

 L101   Why was used V/V, and not w/w? The original published procedures used v/v, not w/w.  For continuity I followed the same designation.

L103   Can authors present schematically this process? It is very important for the clarity of this experiment. I do not feel a schematic drawing depicting the workflow is necessary, especially for this short Communication.  There is adequate detail within the text of this Communication for readers to follow the procedures.  Besides, the published works from which our procedures were taken are cited and readers wishing more detail should go back to that literature.

L137   Please remove italic font. This has been corrected.

L143   This Figure should be with Fig. 1 and Fig. 2, in the context of experimental set up here. I disagree with the reviewer.  There were two distinct experimental setups: one for measuring light conduction in a live plant; and the other for measuring light conduction within suspended suberin solutions. Hence, we ask to leave Figure 5 in its original order.

L180   Please explain the term Casparian strips, what it is made of, and what it is the function, with references. Please see Introduction Lines 79-92.

L188   Please put residencies used for the text in Lines 184-194.  I am not sure what the reviewer means by “residencies.” If the reviewer is referring to light wavelengths, it is clearly stated in the Methods that the halogen lamp emits a full spectrum of light. We did not attempt to use bandpass filters for the visible light range but did perform measurements using a filter that only allowed infrared light to pass through to the sensor. Also note that Figure 1 has an inset with the wavelength light output characteristics of the halogen lamp.

Reviewer 2 Report

The communication entitled “Plants Utilize Suberin Biopolymers as a Vector for Transmitting  Visible Light through their Roots” is a clear work that deals about how light can be transmitted towards roots. The paper describes an interesting approach that deserves attention.

I have some questions and comments, just to improve some minor aspects:

1.- The introduction describes very well, in a nice and academic way, the biological context of the work. However, I miss a paragraph dedicated to the biomimetic of suberin that explains the advantages of using such inspired polymers in this type of studies.

2.- l89. “The other side of the tube”. The sentence is not finished.

3.- What are the limitations, in terms of chemical composition, of using cork suberin instead of maize suberin from roots? I am mainly thinking about the polyaromatic domains of suberin.

4.- l101-102. “…0.5m KOH solution in ethanol/water (2.8 g NaOH dissolved in 90 mL ethanol/water, 9:1 v/v)…”. “0.5m” should be “0.5 M”. Are you using KOH or NaOH (both are described)? In any case, this methodology leads to the opening of the epoxy groups to form the corresponding diols.

5.- Please, add some chemical proofs to check that the polymerization of suberin monomers was successful.

6.- l137. “Data was…”. These lines should not be in italic.

7.- l150. “H. serpedicae” should be in italic.

8.- l156. “Light transmission”. The sentence is not finished. Is something related to the biomimetic suberin polymer full spectrum light conductance? And about the infrared region?

Author Response

Responses to Reviewer 2 are italicized below.

1.- The introduction describes very well, in a nice and academic way, the biological context of the work. However, I miss a paragraph dedicated to the biomimetic of suberin that explains the advantages of using such inspired polymers in this type of studies.  Please see lines 74-91 in the Introduction which provides the rationale that led us to examine light transmission in suberin.

 2.- l89. “The other side of the tube”. The sentence is not finished. This has been corrected.  It seems that formatting by the editorial office resulted in manuscript text getting integrated into a figure caption.

 3.- What are the limitations, in terms of chemical composition, of using cork suberin instead of maize suberin from roots? I am mainly thinking about the polyaromatic domains of suberin.  This is an excellent point.  While we demonstrate light conduction within maize root cell zones that are rich in suberin, we utilized commercial powdered cork as a source of abundant suberin to examine light conduction properties of the co-monomers versus the polymer.  This observation may not extrapolate universally to maize suberin as chemical composition of the biopolymer can vary across tissue types within the same plant and across different plant species.  This was pointed out in lines 87-89 and lines 210-213 with reference 34 cited as a paper detailing this.

 4.- l101-102. “…0.5m KOH solution in ethanol/water (2.8 g NaOH dissolved in 90 mL ethanol/water, 9:1 v/v)…”. “0.5m” should be “0.5 M”. Are you using KOH or NaOH (both are described)? In any case, this methodology leads to the opening of the epoxy groups to form the corresponding diols. NaOH was used in the procedure and the error has been corrected.  I agree that the alkaline hydrolysis of suberin will likely open any epoxide rings.  This was not verified at the time through rigorous GC-MS analysis.

 5.- Please, add some chemical proofs to check that the polymerization of suberin monomers was successful.  We performed a simple melting point determination where results seemed to match with prior published work with cork derived suberin.  We did not perform other more detail analytical workup. However, the different melting temperature ranges between co-monomer and polymer coupled with our observing significantly different light transmission properties should be compelling evidence that our re-polymerization of the co-monomers was successful.

6.- l137. “Data was…”. These lines should not be in italic. This has been corrected – this occurred during editorial office re-formatting.

7.- l150. “H. serpedicae” should be in italic. Corrected

8.- l156. “Light transmission”. The sentence is not finished. Is something related to the biomimetic suberin polymer full spectrum light conductance? And about the infrared region? Again, apologies for this – the missing text ended up in a figure caption during editorial office re-formatting.  This has been corrected.

Round 2

Reviewer 2 Report

All comments have been correctly addressed. 

Congrats to the authors.